# The Effect of Amino Acids on Production of SCFA and bCFA by Members of the Porcine Colonic Microbiota

**DOI:** 10.3390/microorganisms10040762

**Published:** 2022-03-31

**Authors:** Pieter Van den Abbeele, Jonas Ghyselinck, Massimo Marzorati, Anna-Maria Koch, William Lambert, Joris Michiels, Tristan Chalvon-Demersay

**Affiliations:** 1ProDigest, 9052 Ghent, Belgium; jonas.ghyselinck@prodigest.eu (J.G.); massimo.marzorati@prodigest.eu (M.M.); 2Cryptobiotix, 9052 Ghent, Belgium; pieter.vandenabbeele@cryptobiotix.eu (P.V.d.A.); 3Center of Microbial Ecology and Technology (CMET), Ghent University, 9000 Ghent, Belgium; 4METEX NOOVISTAGO, 75017 Paris, France; am.koch@scheerkoch.de (A.-M.K.); william.lambert@metex-noovistago.com (W.L.); 5Laboratory for Animal Nutrition and Animal Product Quality, Ghent University, 9000 Ghent, Belgium; joris.michiels@ugent.be

**Keywords:** amino acids, gut microbiota, bCFA, SCFA, piglet, colon

## Abstract

Functional amino acids supplementation to farm animals is considered to not only be beneficial by regulating intestinal barrier, oxidative stress, and immunity, but potentially also by impacting the gut microbiota. The impact of amino acids on a piglet-derived colonic microbiota was evaluated using a 48-h in vitro batch incubation strategy. The combination of 16S rRNA gene profiling with flow cytometry demonstrated that specific microbial taxa were involved in the fermentation of each of the amino acids resulting in the production of specific metabolites. Branched chain amino acids (leucine, isoleucine, valine) strongly increased branched-chain fatty acids (+23.0 mM) and valerate levels (+3.0 mM), coincided with a marked increase of *Peptostreptococcaceae*. Further, glutamine and glutamate specifically stimulated acetate (~20 mM) and butyrate (~10 mM) production, relating to a stimulation of a range of families containing known butyrate-producing species (*Ruminococcaceae*, *Oscillospiraceae,* and *Christensenellaceae*). Finally, while tryptophan was only fermented to a minor extent, arginine and lysine specifically increased propionate levels (~2 mM), likely produced by *Muribaculaceae* members. Overall, amino acids were thus shown to be selectively utilized by microbes originating from the porcine colonic microbiota, resulting in the production of health-related short-chain fatty acids, thus confirming the prebiotic potential of specific functional amino acids.

## 1. Introduction

Pressure to reduce the use of antibiotics and ZnO urges to find nutritional strategies to mitigate the effect of pathogens on farm animals’ health and performance. Rather than targeting pathogens, nutritional alternatives are believed to support gut health resilience. For example, the effects of functional amino acids on the four different pillars of gut health have been recently reviewed [1]. Functional amino acids supplementation is believed to be beneficial in challenged animals not only by regulating intestinal barrier, oxidative stress, and immunity but also by modulating microbiota which in turn can improve host health. Amino acids can also reach the colon given that crude protein digestibility is only between 70–90% (thus resulting in 10–30% amino acids reaching the large intestine) [2]. These latter effects were firstly described in mice where arginine or glutamine supplementation at a high dose was associated with an increased population of *Lactobacillus* in the jejunum [3,4]. Similarly, supplementation with branched chain amino acids increased the abundance of the *Akkermansia* and *Bifidobacterium* in feces, along with a decreased abundance of *Enterobacteriaceae* which had beneficial effects on the host [5]. The microbiota-modulating effect of AAs has already been investigated and confirmed in farm animals. For example, L-arginine supplementation increases both the *Bacteroidaceae* family and the *Bacteroides* genus in feces of gestating sows [6]. One study investigated the effect of supplementing 3 g of monosodium L-glutamate (MSG) for 30 days on the microbiota from jejunum, ileum, cecum, and colon of growing pigs [7]. This study revealed that MSG supplementation increased the diversity of microbiota which can be regarded as a positive effect. In broilers, it has been reported that L-arginine supplementation can normalize the ileal microbiota of *C. perfringens*-challenged chickens [8]. In addition, in broiler chickens facing a 2-h transportation stress, L-tryptophan supplementation increased the population of beneficial bacteria at the expense of pathogenic ones [9]. Similarly, sulfur amino acid supplementation exerted a beneficial effect on broiler cecal microbial community by increasing the alpha diversity of the microbiota [10]. Taken together, these results suggest that amino acid supplementation could be interesting to modulate positively the microbiota of animals. However, the literature remains scarce and the measurement of amino acid-derived metabolites, which could drive the response on the host health, is usually missing.

While in vivo studies are indispensable to demonstrate a health effect of an intervention, they suffer from the inability to sample at the site of activity unless the animals are sacrificed. Moreover, there is a large inter-individual variation, both in physiological parameters (e.g., gastric emptying and transit times) [11,12] as well as in gut microbiome composition [13]. This interindividual variability generates noise that makes it difficult to identify the mechanism of action of interventions. In vitro models for the porcine gut microbiota range from static batch fermentations [14] up to continuous models making use of fermenters [15] or dynamic models such as the PolyFermS model [16]. The advantage of such models is the higher reproducibility and throughput, especially for static batch fermentation models. When batch incubations are properly designed (i.e., relevant in vivo buffering capacity, no excessive dosage of test product), the limitations of batch fermentation (i.e., accumulation of fermentation products) can be overcome, thus rendering excellent tools to complement in vivo findings.

While many studies applied rRNA gene-based techniques that provide qualitative insights into gut microbial composition (i.e., the proportion of a given group) [13,14,15,16], some recent studies started to combine 16S rRNA gene profiling with an enumeration of total cell numbers in a given sample, thus providing quantitative insights [17,18]. This allows to gain a better understanding of how microbial communities are modulated, particularly during interventions that increase (e.g., fiber) or decrease (e.g., antibiotic) total cell numbers. The synergy of combining sequencing data with flow cytometry has been shown particularly useful to elucidate treatment effects on the gut microbiota during multiple recent in vitro studies [19,20,21].

Hence, this study aimed to investigate the impact of six amino acids on microbial activity and composition of a piglet-derived colonic microbiota using a short-term in vitro incubation method. Changes in microbial composition were investigated upon combining 16S rRNA sequencing with flow cytometry allowing to obtain quantitative insights into community modulation.

## 2. Materials and Methods

### 2.1. Chemicals

All chemicals were obtained from Sigma (Bornem, Belgium) unless otherwise stated. Pure amino acids used during the project, i.e., arginine (Arg), branched-chain amino acids (bCAA—leucine, isoleucine, valine in a 1:1:1 ratio), glutamine (Gln), glutamate (Glu), lysine (Lys), tryptophan (Trp) were provided by METEX NOOVISTAGO (Paris, France). The amino acids were selected based on two criteria (i) being registered in Europe to be used as nutritional additives, and (ii) literature data suggesting that supplementation is associated with beneficial effects on the microbiota as described in the introduction.

### 2.2. Incubation Strategy

Forty-eight-hour incubations were performed to assess the impact of six amino acids on the porcine colonic microbiota and their fermentation products. Each of the treatments was compared to a no-substrate control (blank). As all incubations were performed in technical triplicate and as the experiment was conducted in two runs (each with its own blank), this resulted in a setup with 24 independent reactors (Figure 1A).

An in vitro method to study fermentation by porcine colonic microbes was implemented based on established short-term batch fermentation approaches for studying fermentation by fecal microbiota of human infants [22], human adults [23], but also cats and dogs [19]. All aforementioned publications employ an incubation strategy that uses a specific nutritional medium that has been demonstrated to maintain a broad range of gut microbes. Therefore, this specific medium was also implemented during the current study with the aim of maintaining a broad diversity of porcine colonic microbes. The incubation strategy involved the following steps. First, 63 mL of colonic medium (K_2_HPO_4_, 3.5 g/L; KH_2_PO_4_, 10.9 g/L; NaHCO_3_, 2 g/L; yeast extract, 2 g/L; peptone, 2 g/L; starch, 2 g/L; mucin, 1 g/L; cysteine, 0.5 g/L; and Tween 80, 2 mL/L) was added to 120 mL glass bottles (Novolab, Belgium), containing the correct amount of the different test products to reach a final concentration of 3 g/L test product. Upon the addition of the nutritional medium, glass bottles were sealed with butyl rubber stoppers and flushed with N_2_ to obtain anaerobiosis. Subsequently, 7 mL of a cryopreserved porcine colonic inoculum was administered, after which the incubations were initiated for a period of 48 h. This 48 h duration was applied in consistency with earlier publications with similar incubation strategies [19,22,23]. During this period, the temperature was controlled at 39 °C and continuous mixing was ensured by shaking the reactors at 90 rpm. During the incubation, samples were collected for the analysis of microbial metabolic activity (pH, gas production, SCFA, lactate, bCFA, ammonium, and lactate production) and microbial composition (16S rRNA gene profiling combined with flow cytometry) at 0 h, 24 h, and 48 h (Figure 1B).

The inoculum was derived from a freshly collected sample of the mid-colonic content of a piglet (22.1 kg, 10 weeks of age). The mid-colon refers to a 20-cm section cranial to the end of the centripetal coils of the colon. After weaning, the pig was fed a cereal-soy-based weaner (2 weeks) and starter (4 weeks) diet without antimicrobial growths promoters or probiotics until collection (Table 1 in Van Noten et al., 2020 [24]). After euthanasia, the abdominal cavity of the pig was opened, and the mid-colonic section was clamped with forceps and excised. Its contents were poured into an aseptic container flushed with nitrogen. Immediately hereafter, this sample was mixed in a 15:100 (mass/volume) ratio with anaerobic phosphate buffer (K_2_HPO_4_, 8.8 g/L; KH_2_PO_4_, 6.8 g/L; sodium thioglycolate, 0.1 g/L; and sodium dithionite, 0.015 g/L). After homogenization (10 min, BagMixer 400, Interscience, Louvain-La Neuve, Belgium) and removal of large particles via mild centrifugation (500 g for 2 min), the inoculum was mixed 1:1 (volume/volume) with anaerobic cryoprotectant (prepared according to Hoefman et al., 2013 [25]), aliquoted and preserved at −80 °C (after flash freezing in liquid nitrogen in an anaerobic workstation (TCPS Ltd., Belgium) with gas atmosphere 80/10/10 N_2_/CO_2_/H_2_). Preparation of aliquots from a single colonic suspension ensured that identical microbial communities could be used throughout different project runs which was important given that the project was performed in two runs. While Arg, bCAA, Lys, and Trp were tested during run 1, Gln and Glu were tested during run two. As will be elaborated in the results section, a blank was run during each of the two runs (blank and blank’) and these two blanks appeared to be highly similar in terms of microbial activity and composition, thus allowing to compare treatment effects of amino acids tested in both runs. To optimally represent the exact treatment effects, the results are presented as the difference between a given treatment versus the corresponding blank of a given project run.

### 2.3. Microbial Activity Analysis

Gas production was measured with a pressure meter (Hand-held pressure indicator CPH6200; Wika, Echt, The Netherlands) and pH measurements were carried out with a Senseline pH meter F410 (ProSense, Oosterhout, The Netherlands). pH values were shown as the difference compared to the blank incubation. The pH of the blank at 0 h, 24 h, and 48 h was respectively 6.61 ± 0.01, 6.40 ± 0.01, and 6.34 ± 0.01. Lactate was quantified using a commercially available kit, according to manufacturer’s instructions (R-Biopharm, Darmstadt, Germany). Further, short-chain fatty acids (SCFA; acetate, propionate and butyrate, valerate and caproate) and branched-chain fatty acids levels (bCFA: isobutyrate, isovalerate, and isocaproate) were measured via a GC-FID method described by De Weirdt et al. [26], after applying a diethyl ether extraction (with addition of 2-methyl hexanoic acid as internal standard). Finally, ammonium was quantified via steam distillation, followed by titrimetric determination with HCl [27].

### 2.4. Microbial Composition Analysis

DNA was extracted from the pellet of 1 mL sample (obtained upon centrifugation during 5 min at 9000 g) as described by Boon et al. [28] with modifications implemented by Duysburgh et al. [29]. 16S-rRNA gene profiling was performed by LGC Genomics GmbH (Berlin, Germany) as described recently [20]. As elaborated in [30,31], the analysis was adapted from the MiSeq protocol for read assembly and cleanup using the mothur software (v. 1.39.5) as follows. First, reads were assembled into contigs, followed by alignment-based quality filtering. Upon removing chimeras, taxonomy was assigned via a naïve Bayesian classifier [32] and RDP release 14 [33]. Finally, contigs were clustered into OTUs at 97% sequence similarity. All sequences that were classified as Eukaryota, Archaea, Chloroplasts, and Mitochondria were removed. Also, if sequences could not be classified at all (even at (super)Kingdom level) they were removed. The final number of combined reads was on average 28,858 (minimum 11,580; maximum 50,132). Rarefaction curves were made with Past 4.03 [34] to confirm that the sequencing depth allowed to grasp the microbial diversity of samples of the porcine inocula and in vitro samples. The proportional phylogenetic information (%) was combined with a quantification of the total number of cells via flow cytometry. Pertaining to the latter, as described by Van den Abbeele et al. [20], samples were first diluted in Dulbecco’s Phosphate-buffered Saline (DPBS) after which they were stained with 0.01 mM SYTO24 (Life Technologies Europe, Merelbeke, Belgium), prior to being run on a BD Facsverse (BDBiosciences, Erembodegem, Belgium) and analyzed using FlowJo, version 10.5.2. It should be noted that the data obtained by multiplying sequencing data (%) with cell numbers derived from flow cytometry (cells/mL) results in an absolute amount of a given taxonomic group. This amount should be considered as an estimated amount (cells/mL) given that the sequencing analysis also only provides an estimated abundance. One microbial cell can for instance have multiple copies of the 16S rRNA gene, while the number of copies can also differ between microbial species.

### 2.5. Statistics

For exploratory data analysis, Principal Component Analysis (PCA) was performed for both metabolic and compositional data (at family level) via GraphPad Prism version 9.2.0 (Graphpad Software, San Diego, CA, USA). The averages across technical replicates were calculated and 2-sided t-tests were performed to identify significant effects as opposed to the corresponding blank. As there were six simultaneous comparisons (blank versus each of the six amino acids), multiplicity was corrected using the Benjamini-Hochberg false discovery rate (FDR, with FDR = 0.05 for both metabolic markers and 16S rRNA gene profiling) [35]. All calculations were carried out in Excel, while figures were prepared in GraphPad Prism version 9.2.0 (Graphpad Software, San Diego, CA, USA).

Regularized Canonical Correlation Analysis (rCCA) was done to highlight correlations between the metabolic and compositional data (at family level). Regarding compositional data, proportional phylogenetic data was used as input. rCCA was executed using the mixOmics package with the shrinkage method for estimation of penalisation parameters (version 6.16.3, Rohart et al., 2017) in R (https://www.r-project.org/, accessed on 13 March 2022) [36].

## 3. Results

### 3.1. Amino Acids Differentially Stimulated Microbial Activity of the Porcine Colonic Microbiota

As elaborated in the materials and methods, the project was performed in two runs, each run using the same cryopreserved porcine colonic inoculum. The cryopreservation method used to store the porcine colonic inoculum allowed for reproducible incubations as followed from the similar levels of various markers of microbial activity for the two independent blank incubations (blank and blank’) (Appendix A), resulting in the colocalization of both blank samples in the overall PCA (Appendix A). This enabled the direct comparison of treatment effects on microbial activity of the various amino acids tested along the project.

To gain insight into overall changes in microbial activity, the data was first presented in a PCA (Figure 2A), which explained a great part of the overall variation (cumulative variation explained by PC1/PC2 = 75.1%), thus stressing that the PCA provided optimal insight in the factors underlying the variation. While the first factor was time (visualized by colocalization of samples of the three time points (0 h, 24 h and 48 h), mostly along PC1), a second factor was the treatment with the different amino acids (visualized by the more distant positioning of treatment samples versus the untreated blank as time progressed (from 0 h to 48 h)).

The levels of the different markers of microbial activity provided insights into how the amino acids differentially modulated microbial activity (Figure 3 and Figure 4). Given that both figures show the difference between the treatment and the corresponding blank, positive and negative values suggest respectively enhanced and lowered production of a given metabolite compared to the blank. Notably, Arg resulted in more alkalinity of the medium and Glu in more acidity, while other amino acids minimally affected pH as compared to a blank (Figure 3A). Upon their administration, Arg and Glu instantaneously increased and decreased the pH from 6.54 to 6.80 and 6.16, respectively. These differences in pH were thus maintained throughout the incubation. Gas production increased for all amino acids after 48 h of incubation, however with large differences among amino acids (Figure 3B). Further, while all amino acids increased ammonium production at 48 h as compared to blank, suggesting fermentation of each amino acid, the highest increases were noted for Arg/Gln, followed by bCAA/Glu and Lys/Trp (Figure 3C). Further, a specific and substantial increase of bCFA production was noted upon bCAA treatment due to increases in isobutyrate, isovalerate, and isocaproate (Figure 3D–F). bCAA additionally increased levels of valerate and caproate (Figure 4E,F). In contrast, Gln and Glu strongly and specifically increased acetate and butyrate levels (Figure 4A,C), coinciding with most marked increases in gas production. Gln and Glu also mildly increased caproate levels. Arg and Lys specifically increased propionate production (Figure 4B). Lactate could only be quantified in 24 h incubations (Appendix A), with either reductions (bCAA, Gln) or elevations (Glu, Lys) when compared to the blank (Figure 4D). Finally, Trp hardly impacted markers of microbial activity except for a minor increase in valerate levels. As a remark, in some cases, the stimulation of bCFA/SCFA production coincided with lower levels of other metabolites, e.g., bCAA decreased acetate and propionate levels at 48 h.

### 3.2. Amino Acids Differentially Altered Microbial Composition of the Porcine Colonic Microbiota

Analysis of the microbial composition via quantitative 16S rRNA gene profiling confirmed that the cryopreservation method used to store the porcine colonic inoculum allowed for reproducible incubations. This conclusion was based on the detection of similar levels of various taxonomic groups in the two independent blank incubations (blank and blank’) (Appendix A and Appendix A), resulting in colocalization of both blanks in the overall PCA (Appendix A). This high reproducibility enabled the direct comparison of treatment effects on the microbial composition of the various amino acids tested along the project.

To gain insight into treatment effects on microbial composition, the data was first presented at the phylum level, both as proportional (Figure 5A) and absolute levels (Figure 5B). Besides revealing some high-level product-specific effects (e.g., stimulation of *Bacteroidetes* by Arg versus the untreated blank, supported by statistical analysis (Appendix A)), this also demonstrated that the various amino acids differentially affected the total number of microbial cells, suggesting that proportional numbers could obscure part of the true treatment effects on microbial composition. Therefore, in consistency with recent publications [19,20,21], a subsequent analysis of microbial composition was performed based on estimated absolute levels.

Like for microbial activity, the PCA based on microbial composition (at family level) explained a great part of the variation of the dataset (68%) (Figure 6A). Besides a clear impact of time (visualized by colocalization of samples of the three time points (0 h, 24 h and 48 h)), a second factor determining the positioning of samples in the PCA was the treatment with specific amino acids. Treatment effects were again most pronounced at 48 h as visualized by a more distant positioning of treatment samples *versus* the untreated blank as time progressed. In other words, at 24 but especially at 48 h, samples of the different treatments were positioned more distant from the untreated blank. Apparently, Arg exhibited a highly different microbial community.

The exact absolute levels of the families underlying this clustering provided insights into how the amino acids differentially modulated microbial composition at 24 h (Appendix A) and 48 h (Table 1). At 48 h, Arg most markedly altered microbial composition in comparison to the untreated blank. Arg particularly stimulated *Eggerthellaceae*, *Muribaculaceae,* and *Tannerellaceae*, while also *Enterococcaceae* tended to increase (*p* = 0.055) (Table 1). Further, bCAA and Lys specifically stimulated *Peptostreptococcaceae* and *Muribaculaceae*, respectively. The specific increase of *Peptostreptococcaceae* with BCAA was due to the stimulation of an OTU related to *Peptostreptococcus russellii.* The similar modulation of microbial activity by Gln and Gln was in line with similar increases in *Ruminococcaceae*, *Oscilospiraceae,* and *Acidaminococcaceae*, all within phylum Firmicutes. Nonetheless, Glu specifically increased *Enterobacteriaceae* and *Paludibacteraceae*, Gln specifically increased *Christensenellaceae*, *Muribaculaceae,* and *Tannerellaceae*. Finally, Trp mostly stimulated *Enterobacteriaceae* and *Peptostreptococcaceae* at the expense of *Tannerellaceae*.

### 3.3. Functional Populations Involved in the Fermentation of Amino Acids

To uncover potential functional groups, an rCCA with the metabolic and compositional data (at family level) at 48 h of incubation (Figure 7). This highlighted the strong correlation between the presence of *Peptostreptococcaceae* and the production of valerate and bCFA (isobutyrate, isovalerate, and isocaproate). Further, *Muribaculaceae* and *Tannerellaceae* strongly correlated with propionate production. Finally, *Acidaminococcaceae*, *Oscillospiraceae*, *Rikenellaceae,* and *Christensenellaceae are* most markedly related to high acetate and butyrate levels (amongst other families that were not affected by any of the amino acids).

### 3.4. In Vitro Microbiota Composition in Comparison with the Original Porcine Colonic Inoculum

While a broad diversity of microbes was maintained throughout the 48 h incubations it was also noted that specific taxonomic groups were underrepresented during the in vitro incubations. This was reflected by a clustering of these families together with the original inoculum (0 h; left side of the PCA in Figure 6B). Overall, the following families tended to be underrepresented during in vitro incubations: *Paludibacteraceae*, *Prevotellaceae*, *Rikenellaceae*, *Butyricicoccaceae*, *Christensenellaceae*, *Oscillospirales,* and *Spirochaetaceae.*

## 4. Discussion

The strength of the applied methodology resided in the high reproducibility of the incubation strategy together with the accurate method used to investigate microbial composition (16S rRNA gene profiling combined with flow cytometry). This allowed to demonstrate that specific microbial taxa were involved in the fermentation of the various amino acids resulting in the production of specific SCFA/bCFA. Obtaining such insights is challenging during in vivo studies given the intrinsic variation during such studies while metabolites such as SCFA are rapidly absorbed after being produced [37]. When comparing the production of ammonium (NH_4_^+^) with the theoretical total nitrogen content of each of the amino acids, it followed that 88%, 90%, and 81% of nitrogen was converted to ammonium for bCAA, Gln, and Glu, respectively. The high bCFA/SCFA production for these three AAs further confirmed an almost complete fermentation of bCAA, Gln, and Glu. Further, while nitrogen conversion to ammonium was intermediate for Arg (53%), it was low for Trp (15%) and Lys (8%) corresponding with intermediate and minor changes in bCFA/SCFA production for these AA, respectively. Overall, specific changes in microbial activity for each of the AA related to specific changes in the abundance of microbial taxa. Our findings thus highlight the prebiotic potential of amino acids and stress that the concept of prebiotics could go beyond the use of indigestible carbohydrates such as the gold-standard prebiotic inulin [38]. Indeed, according to a recent international scientific consensus prebiotics are now defined as substrates that are selectively utilized by host microorganisms, thus conferring a health benefit [39].

First, bCAA administration strongly increased the production of bCFA (+23.0 mM) and valerate (+3.0 mM), and to a lesser extent also caproate (+0.3 mM). These changes coincided with a marked increase of *Peptostreptococcaceae*, at a lower taxonomic level due to a specific increase of an OTU related to *Peptostreptococcus russellii*. As this species was originally isolated from a swine-manure storage pit using an isolation procedure tweaked for the enrichment of peptide and amino acid-fermenting bacteria [40], *Peptostreptococcus russellii* is likely a key BCAA-fermenting species within the porcine gut microbiota. Interestingly, while valerate is much less studied than the other SCFA, it has been attributed to health-related effects such as decreasing the growth of cancer cells [41] or exerting antipathogenic effects against *C. difficile* [42]. Further, while also little is known about the impact of bCFA on host health, there is some evidence that bCFA can be oxidized when butyrate is not available, thereby contributing to health benefits [43]. In addition, a study has shown that a mix of bCFA (isobutyrate and isovalerate) was able to prevent gut permeability induced by pro-inflammatory cytokines in the Caco-2 cell line model [44]. bCAA also increased butyrate levels (+2.3 mM), while decreasing acetate (−9.3 mM) and propionate levels (−1.7 mM). While the decrease in acetate levels could in part be due to its conversion to butyrate [45], part of the decrease could have resulted from an inhibition of acetate and propionate-producing gut microbes. Nevertheless, pronounced stimulation of other metabolites stress the potential of BCAA to exert potential health benefits upon its fermentation by porcine gut microbes.

Glu and Gln resulted in a completely different spectrum of microbial metabolites as they mostly stimulated the production of acetate (~20 mM) and butyrate (~10 mM). While both AA increased the levels of *Ruminococcaceae*, *Oscillospiraceae,* and *Acidaminococcaceae*, Gln additionally increased *Christensenellaceae.* The correlation analysis suggested the involvement of all four families in the production of acetate and butyrate. According to Buckel et al. (2001) who reviewed different pathways for Glu conversion key pathways are likely (i) a coenzyme B_12_-dependent glutamate mutase-mediated pathway resulting in the formation of ammonium, acetate, and pyruvate (further converted to CO_2_ and butyrate), and (ii) a pathway involving decarboxylation of glutamate leading to 4-aminobutyrate, which is fermented by a second reaction to acetate and butyrate (via dehydration of 4-hydroxybutyryl-CoA to crotonyl-CoA) [46]. While there can be a large variation in functional properties of species belonging to the same family, the three of the aforementioned families that increased upon Glu/Gln supplementation indeed contain members capable of producing butyrate. These families include the *Ruminococcaceae* family [47], while also the *Oscillospiraceae* and *Christensenellaceae* families contain known butyrate producers, even considered to be potential next-generation probiotics [48,49]. Given the well-described health benefits of both SCFA as reviewed by Rivière et al. [50], consumption of Glu and Gln could result in beneficial effects on host health via fermentation by the porcine gut microbiota.

Fermentation of Arg and Lys resulted in a specific increase in propionate levels (~2 mM). Interestingly, both AA increased the abundance of *Muribaculaceae,* a family that is underreported in literature given that classification using the Ribosomal Database Project assigns sequences to the *Porphyromonadaceae* instead [51]. During the current project, its abundance strongly correlated to propionate production. Genomes assembled from metagenomes suggest that *Muribaculaceae* members are capable of producing propionate, while occupying a similar niche as *Bacteroidaceae* [52]. Interestingly, increases in *Bacteroidaceae* and propionate levels have already been observed upon Arg supplementation to gestating sows [6], further suggesting the involvement of *Bacteroidetes* members in the fermentation of Arg and Lys. Given the well-described health benefits of propionate as reviewed by Hosseini et al. [53], consumption of Arg and Lys could result in beneficial effects via fermentation by the gut microbiota.

While both NH_4_^+^ and bCFA are reliable markers for protein fermentation, the current study stresses key differences between both markers. While NH_4_^+^ is produced upon fermentation of all AA (with higher levels being produced upon fermentation of AAs that contain more N-atoms), bCFA are exclusively produced upon fermentation of bCAA (leucine, isoleucine, valine). Then, while SCFA are often considered markers of saccharolytic fermentation, the current study also stresses that each of the SCFA (especially acetate and butyrate) are also produced upon amino acid fermentation.

While the current study thus focused on ammonia, SCFA, and bCFA as endpoints of protein fermentation, AA can also be converted in more toxic compounds such as phenols and sulfides [54]. On the other hand, metabolites of AA have also been linked to the gut-brain axis [55]. Future studies could thus additionally target such metabolites to get a broader understanding of AA on health. Finally, a limitation of the current study was that despite that a wide range of families derived from the original inoculum was cultivated and detected at 24 h and 48 h, a considerable amount of taxa were more abundant in the inoculum as opposed to the in vitro samples, suggesting that the applied in vitro conditions can be further optimized. A marked difference was noted within the Bacteroidetes phylum where *Muribaculaceae*, *Prevotellaceae*, *Paludibacteraceae*, *Rikenellaceae,* and *Tannerellaceae* families were the key families in the inoculum. In contrast, the *Bacteroidaceae* family was almost exclusively representing the *Bacteroidetes* phylum after 48 h of incubation. Amongst others, these taxa should thus deserve more attention during future model development. In part, the limited outgrowth of some of the microbial taxa could be because the inoculum was cryopreserved prior to its use. While this offered the advantage of being able to perform reproducible tests on different occasions, it cannot be excluded that some taxa suffered from these preservation procedures. While it was outside the scope of this project, future studies could further investigate the potential differences between experiments done with fresh and cryopreserved samples. Finally, this study investigated the fermentation of amino acids by the microbiota-derived from a single piglet. Given the importance of interindividual differences across different animals, it would be interesting to confirm the findings of the current study for a broader range of animals to demonstrate that the current findings are not restricted to the specific microbiota under investigation but are truly representative of a broader cohort.

## 5. Conclusions

Overall, this study demonstrated that the different amino acids, especially BCAA, Gln, Glu, and Arg are selectively utilized by microbes originating from the porcine colonic microbiota. This resulted in the production of health-related metabolites such as acetate, propionate, butyrate, and/or valerate. These findings thus highlight the prebiotic potential of amino acids and stress that the prebiotic concept could go beyond the use of indigestible carbohydrates. While the aspect of selective utilization by host microorganisms (criterium one of prebiotic definition) is clearly demonstrated during the current study, subsequent health benefits (criterium two of prebiotic definition) are likely given the production of health-related SCFA upon AA fermentation.

## Figures and Tables

**Figure 1 microorganisms-10-00762-f001:**
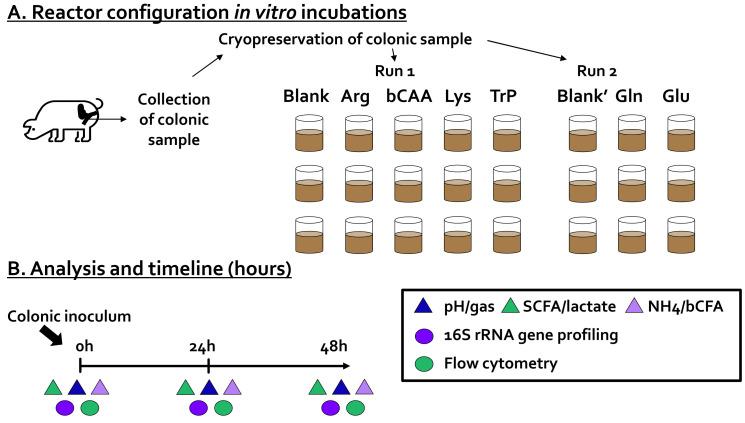
(**A**) Reactor configuration and (**B**) timeline of the in vitro experiment during which the impact of amino acids on the activity and composition of a porcine colonic microbiota was investigated. Arg = arginine, bCAA = branched-chain amino acids (leucine, valine and isoleucine in a 1:1:1 ratio), bCFA = branched-chain fatty acids, Gln = glutamine, Glu = glutamate, Lys = lysine, NH_4_ = ammonium, SCFA = short-chain fatty acids, Trp = tryptophan.

**Figure 2 microorganisms-10-00762-f002:**
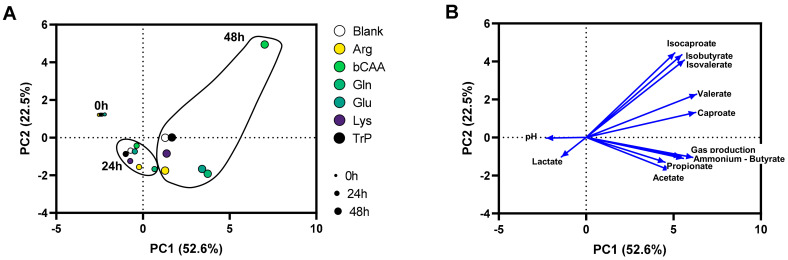
(**A**) PCA based on microbial metabolic activity at different time points (0 h, 24 h, and 48 h) during the incubation of porcine colonic microbiota in presence of various amino acids versus an untreated control incubation (blank). (**B**) Loadings of the parameters on which the PCA was based: pH, gas production, and levels of SCFA (acetate, propionate, butyrate, valerate, caproate), bCFA (isobutyrate, isovalerate, isocaproate), lactate and ammonium. PCA = principal component analysis.

**Figure 3 microorganisms-10-00762-f003:**
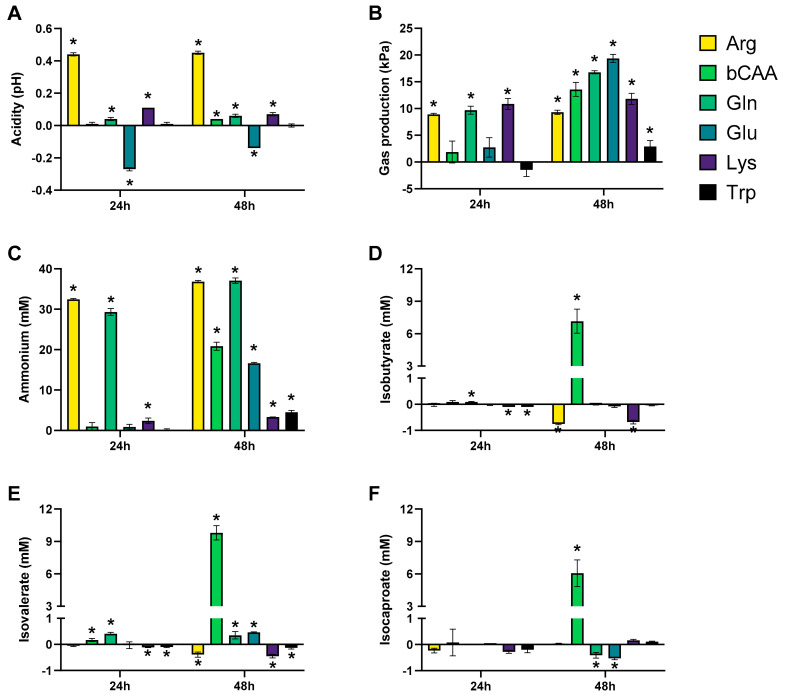
Effect of amino acids on acidity (pH) (**A**), gas production (kPa) (**B**), and markers of proteolytic fermentation (ammonium (**C**) and bCFA (isobutyrate (**D**), isovalerate (**E**), and isocaproate (**F**))) upon incubation with a porcine colonic microbiota (*n* = 3). The data is presented as the average (±SD) difference versus a corresponding blank incubation. Statistically significant differences are indicated with asterisks (* *p* < 0.05).

**Figure 4 microorganisms-10-00762-f004:**
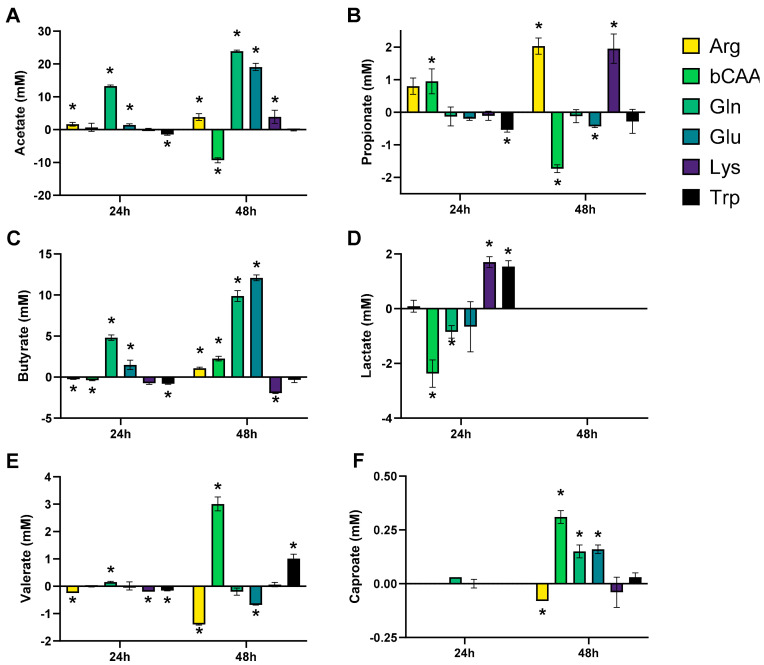
Effect of amino acids on microbial fermentation products (SCFA (acetate (**A**), propionate (**B**), butyrate (**C**), valerate (**E**), and caproate (**F**)) and lactate (**D**)) upon incubation with a porcine colonic microbiota (*n* = 3). The data is presented as the average (±SD) difference versus a corresponding blank incubation. Statistically significant differences are indicated with asterisks (* *p* < 0.05).

**Figure 5 microorganisms-10-00762-f005:**
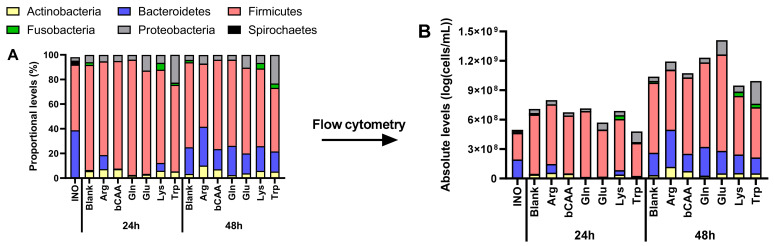
Impact of amino acids on microbial community composition (phylum level) during incubations with a porcine colonic microbiota. Values are the average of three technical replicates (*n* = 3) and are expressed both (**A**) as proportional values based on 16S rRNA gene profiling (%), and (B) as absolute values, estimated upon multiplying with total cell counts (cells/mL) (**B**). INO = inoculum.

**Figure 6 microorganisms-10-00762-f006:**
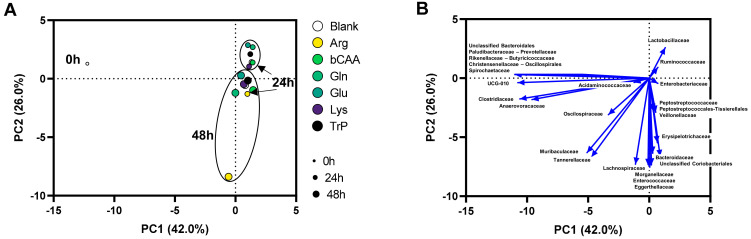
(**A**) PCA based on microbial community composition (family level; absolute values, estimated upon multiplying with total cell counts (cells/mL)) at different time points (0 h, 24 h, and 48 h) during the incubation of porcine colonic microbiota in presence of various amino acids *versus* an untreated control (blank). (**B**) Loadings of the families on which the PCA was based.

**Figure 7 microorganisms-10-00762-f007:**
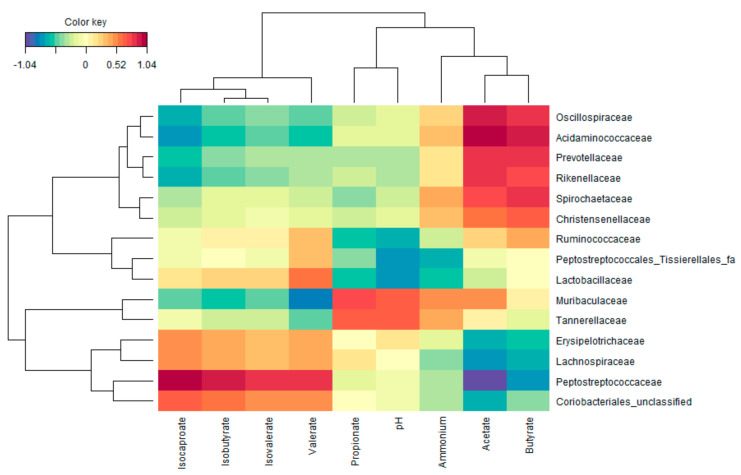
Relationship between bacterial families and pH, gas production, and fermentation products (SCFA, bCFA, and ammonium) in samples collected at the end of the 48 h incubation of the various amino acids with a porcine colonic microbiota. The heatmap was based on the regularized canonical correlation analysis (cutoff = 0.6) between bacterial abundances at family level (proportionate values,) and markers of microbial activity.

**Table 1 microorganisms-10-00762-t001:** Effect of amino acids on microbial composition (family level; absolute values, estimated upon multiplying proportions (%) based on 16S rRNA gene profiling with total cell counts) at the end of the 48 h incubation with a porcine colonic microbiota (*n* = 3). The data is presented as the average difference between the log_10_-transformed absolute abundance (log(cells/mL)) in a treatment versus the corresponding untreated blank incubation. A value below zero indicates a decrease upon treatment, while a value above zero, indicates that this family is stimulated by a given amino acid. Statistically significant differences as compared to this blank are indicated in bold (*p* < 0.05).

Phylum	Family	Arg	bCAA	Gln	Glu	Lys	Trp
Actinobacteria	Unclassified *Coriobacteriales*	**0.61**	**0.46**	−0.18	0.03	0.32	**0.32**
	*Eggerthellaceae*	**1.30**	**0.59**	0.17	0.04	−0.08	0.07
Bacteroidetes	*Bacteroidaceae*	0.27	0.00	0.01	−0.09	0.01	−0.03
	*Unclassified Bacteroidales*	0.22	−0.03	0.06	−0.05	−0.03	−0.03
	*Muribaculaceae*	**2.17**	−0.14	**0.82**	0.04	**1.48**	**−0.43**
	*Paludibacteraceae*	−0.14	−0.34	0.27	**0.59**	−0.24	−0.34
	*Prevotellaceae*	0.11	−0.24	0.32	0.20	−0.08	−0.32
	*Rikenellaceae*	−0.06	−0.53	0.26	0.10	−0.23	**−0.48**
	*Tannerellaceae*	**0.93**	**−0.63**	**0.34**	**−0.34**	0.12	**−1.35**
Firmicutes	*Acidaminococcaceae*	0.24	**−0.60**	**1.07**	**1.27**	0.14	−0.46
	*Anaerovoracaceae*	0.40	−0.18	0.46	0.22	−0.01	0.51
	*Butyricicoccaceae*	0.07	−0.16	0.10	0.35	−0.28	0.06
	*Christensenellaceae*	0.26	−0.08	**0.13**	−0.10	−0.01	**−0.46**
	*Clostridiaceae*	**0.29**	**−0.34**	0.08	0.29	−0.20	−0.16
	*Enterococcaceae*	1.54	0.00	−0.16	−0.44	−0.30	−0.12
	*Erysipelotrichaceae*	**0.25**	−0.01	−0.01	**−1.04**	−0.23	**−0.25**
	*Lachnospiraceae*	**0.26**	−0.06	−0.02	**−0.39**	−0.03	**−0.15**
	*Lactobacillaceae*	**−0.55**	−0.13	−0.09	0.02	−0.26	0.05
	*Oscillospiraceae*	**0.44**	**−0.21**	**0.49**	**0.52**	−0.04	0.05
	*Oscillospirales*	0.37	−0.14	0.25	−0.19	−0.14	−0.02
	*Peptostreptococcaceae*	**0.60**	**1.18**	−0.43	−0.71	**0.33**	**0.61**
	*Peptostreptococcales−Tissierellales*	**−0.36**	**−0.26**	−0.11	**−0.22**	−0.14	−0.02
	*Ruminococcaceae*	**−0.26**	**−0.14**	**0.26**	**0.44**	**−0.23**	**−0.16**
	*UCG−010*	−0.03	**−0.40**	−0.07	**−0.60**	0.09	**−0.37**
	*Veillonellaceae*	**−0.24**	0.00	−0.03	0.06	**0.07**	**−0.52**
Proteobacteria	*Enterobacteriaceae*	**0.27**	0.02	0.00	**0.52**	**0.22**	**0.79**
	*Morganellaceae*	0.73	0.52	−0.49	−1.03	−0.17	0.43
Spirochaetes	*Spirochaetaceae*	0.35	−0.03	0.14	0.07	−0.06	−0.33

## Data Availability

Not applicable.

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
