# Peer review of "The Effect of Amino Acids on Production of SCFA and bCFA by Members of the Porcine Colonic Microbiota"

_microorganisms, 2022, doi:10.3390/microorganisms10040762_

Round 1

Reviewer 1 Report

The authors explored the impact of six amino acids on the metabolic activity of porcine microbiota with nice potential, while the methodologies still need to be further specified and the study design can be improved.

Comments:

  1. The incubation lasted for 48h in vitro. What is the transit time normally for piglet? In other words, how relevant it is to study the incubation for 48h?
  2. To the best of my knowledge, a large number/portion of amino acids can be absorbed already in the upper gastrointestinal tract, could the author add more information about the availability of amino acids to lower gastrointestinal tract (in the introduction)? Like percentage of amino acids remained to the intestinal microbiota after oral ingestion.
  3. Throughout the manuscript, there are many places need to make the text italic, for instance in vivo, in vitro. Please go through the manuscript and correct them accordingly.
  4. For what reason, the six amino acids were selected to be investigated in the current study? This information is still missing in the introduction.
  5. What is the purity of used amino acids in the current study? High purity? if not, please specify and discuss possible impact on the result.
  6. As the study were conducted in two run, do you see the “run” impact on the result? For example, did the blank behave the same in these two runs?
  7. In the method section, the authors mentioned, penicillin bottles were used in the study. The word “penicillin” is quite alarming. Could the author give more information about the “penicillin bottle”?
  8. The test product concentration was set as “3 g/L”. could the author explain how did you decide on this concentration to be used in the current study?
  9. As for the inoculum, the mid-colonic content was collected. Could the author specify, what do you mean by “mid-colonic content”? and how the sample were collected from the piglet?
  10. As also the author stated in the manuscript, the intestinal microbiota demonstrated a great individuality, however, in the current study, only inoculum from one piglet were used. How representative it will be for the general piglet population?
  11. Where did the authors perform the preservation step (for the inoculum)? In an anaerobic chamber? And what the gas inside the anaerobic chamber? These information are necessary in the M&M.
  12. In the result section, the author stated something explained 75.1% of overall variation. To which factor you were trying to point to?
  13. How much variation in the overall metabolites data can be explained by incubation time? By supplemented amino acids?
  14. Figure 3A, the pH changes were expressed as plus or minus zero. What are the original pH value of the medium?
  15. Figure 4, quite a few data showed negative values for acetate, propionate, lactate, valerate etc. how should we interpret the negative values?
  16. In terms of samples collected at 0h (beginning of the incubation), there was only one sample throughout the study. Which sample did the author use as the t0 sample? Can it represent all t0 samples of different treatment?

Author Response

Find attached a document with the original questions, followed by our answers in red/bold. We referred to the manuscript where we applied track changes to facilitate observation of the changes we introduced.

Reviewer 2 Report

The manuscript entitled “Amino acids differentially modulate SCFA and bCFA production by specific members of the porcine colonic microbiota in vitro” is interesting and deals with the Investigation of the impact of six amino acids on microbial activity and composition of a piglet-derived colonic microbiota using a short-term in vitro incubation method. Authors clearly demonstrated the Changes in microbial composition and investigated upon combining 16S rRNA sequencing with flow cytometry allowing obtaining quantitative insights in community modulation and likely to signify the production of health-related SCFA upon AA fermentation.

Author Response

We thank reviewer 2 for appreciating the various aspects of our manuscript.

Reviewer 3 Report

The manuscript entitled "Amino acids differentially modulate SCFA and bCFA production by specific members of the porcine colonic microbiota in vitro" aims to describe the fermentability of specific amino acids by porcine microbiota in an in vitro system.

The manuscript is well written and logically describes the results. The research uses a sound methodology, and the discussion of the results by the authors offers the perspective of amino acids having prebiotic functionality. It is appreciated that the authors also included the limitations of the study including the fact that the validity of the study would be strengthened by the use of colonic content from different animals.

I suggest that the authors include the following information in the methodology section:

1) how was the concentration of the amino acids (3 g/L) were chosen?

2) why the branched amino acids were supplemented as a group as opposed to individually? How would the results differ if they were added individually?

Results section: the authors focus on the increase of SCFA/BCFA, ammonium and lactate production relative to blank.  It is clear, based on the results in figures 3 and 4, that inhibition of SCFA/BCFA and lactate is observed by certain amino acids.  For instance, bCAA significantly inhibited the production of acetate, propionate, and lactate.  What is the implication of the inhibition?  It would be interesting to add this to the discussion. 

Author Response

(The authors gave the same response as above.)

Reviewer 4 Report

The “Amino acids differentially modulate SCFA and bCFA production by specific members of the porcine colonic microbiota in vitro” manuscript does an excellent job demonstrating significant beneficial effect of amino acids supplementation to animals on their health status as well as intestinal microbiota health, which is a subject of great interest among many researchers nowadays.

Since a lot of interest is given to different dietary sources which could be able to fight against intestinal pathogens, due to antibiotics ban, this manuscript showed that some amino acids supplementation could be beneficial in animals not only by regulating intestinal barrier, oxidative stress and immunity but also by modulating microbiota which in turn can improve the overall animal health.

This work is a good reminder for all animal nutritionist to pay attention to the nutritional requirements in essential amino acids from diets, as well as to their link with SCFA and bCFA. I was very pleased to see that the authors were able to obtain quantitative insights in intestinal community modulation.

The authors further presented very well the changes of microbial activity and the insights in how the tested amino acids differentially modulated microbial activity which are showed very clearly in the obtained results. However, it is also interesting to know if this strategy could help the piglets in weaning crisis, because this aspect represents a big problem in the farming industry.

After reading the manuscript, I have few questions for the authors:

  1. Why did they choose these amino acids? What was the criteria for choosing only the six presented AA?
  2. I was a little disappointed that the incubations were performed only in triplicate. Statistically speaking, the number of determinations is a bit small. However, as the authors stated, this study investigated the fermentation of amino acids by the microbiota derived from a single piglet. This leads to the next question. The intestinal activity is the same regarding the age or the hybrid of the piglet from which the porcine colonic samples are collected?
  3. Figure 7 is a bit crowded and the legend is difficult to distinguish. I suggest to be replaced with a better image resolution.
  4. Ethical consideration is missing, regarding the use of animals for experimental purposes.
  5. Why in the conclusion references are given? I suggest drawing a conclusion based on the results obtained in this study. References can be moved to the final paragraph of the discussion chapter.

Author Response

(The authors gave the same response as above.)

Round 2

Reviewer 1 Report

Nicely the authors adapt the manuscript accordingly with additional analysis and acknowledging its limitations in the discussion.